# ENRICHING CONVNETS WITH PRE-CORTICAL PROCESSING ENHANCES ALIGNMENT WITH HUMAN BRAIN RESPONSES

**Niklas Müller, H. Steven Scholte**
Psychology Research Institute
University of Amsterdam
Amsterdam, The Netherlands
{n.muller,h.s.scholte}@uva.nl

**Iris I. A. Groen**
Informatics Institute
University of Amsterdam
Amsterdam, The Netherlands
i.i.a.groen@uva.nl

## ABSTRACT

Convolutional Neural Networks (ConvNets) are the current state-of-the-art for modelling human visual processing whilst also performing tasks on a human performance level. Convolutional features can be seen as analogous to visual receptive fields and thus render them biologically plausible. However, spatially-uniform sampling and reuse of features across the entire visual field do not accurately represent structural properties of the human visual system. Here, we present empirical findings of incorporating functional and structural properties of the human retina into ConvNets on their alignment with human brain activity. We show that predictions of human EEG data using ConvNets features improve by using foveated stimuli and that differential spatial sampling in ConvNets explains several qualities of EEG recordings. We also find that color and contrast filtering of inputs in turn do not yield improved predictions. Overall, our results suggest that incorporating biologically plausible spatial sampling is important for increasing representational alignment between ConvNets and humans.

## 1 INTRODUCTION

Deep neural networks are currently state of the art predictive models of human brain activity and behaviour (Güçlü & van Gerven, 2015; Khaligh-Razavi & Kriegeskorte, 2014). The kernels of a trained convolutional neural network (ConvNet) resemble parts of the computational and representational principles of both early visual cortex, i.e. simple and complex cells in V1, and higher-level visual cortex, in terms of object representations (Lindsay, 2021). However, the performances in predicting human neuroimaging data using features of ConvNets are still below the theoretical maximum of explainable variance (Cichy et al., 2016; Gifford et al., 2022; Huang et al., 2017; Wen et al., 2018), indicating potential to improve their alignment with humans.

In this paper, we argue that certain, built-in features of ConvNets are limiting their accuracy and plausibility as models to study the neural mechanisms underlying early human visual processing. One limitation comes from the spatially uniform input sampling. This architectural feature led to a tremendous success of ConvNets in solving computer vision tasks (Krizhevsky et al., 2012; He et al., 2016; Milletari et al., 2016; Mascarenhas & Agarwal, 2021; Kriegeskorte, 2015; VanRullen, 2017) as reusing convolutional features across the entire visual field reduced training parameters. While this decreased computational resources, it also introduced a gap between the physiology of ConvNets and that of the human retina.

The human retina is structurally divided into areas with a high density of retinal ganglion cells (RGCs) at the center of gaze called fovea, and areas with a lower density of RGCs, referred to as periphery (Oyster et al., 1981; Wässle et al., 1989; 1990; Kreiman, 2021). This distinction is relevant in the context of object recognition. To perceive small object details, humans move their center of gaze, i.e., the fovea, in order to sample visual information about the object with high resolution. Peripheral vision is important in situations when a lot of new visual information needs to be processed, such as turning around to a new scene. While not having high acuity, the periphery

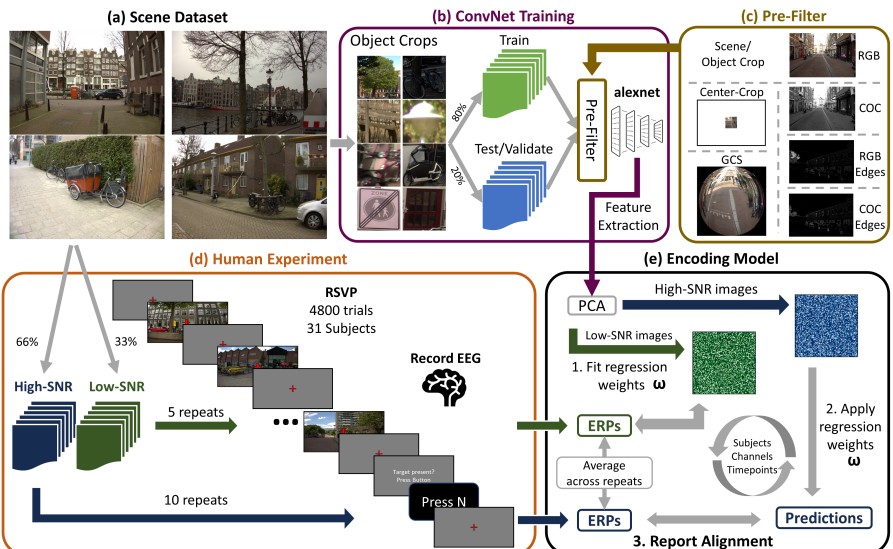

Figure 1: **Methodology (a)** High-resolution outdoor scene examples used for human experiment and ConvNet feature extraction. **(b)** Examples of cropped object images and illustration of train, test, and validation split for alexnet training. **(c)** Pre-filters used to augment ConvNets including different spatial sampling techniques and alternative color coding or contrast filtering. **(d)** Illustration of dataset split for high and low-SNR sets and the RSVP target-detection paradigm while recording EEG. **(e)** Schematic explanation of linearized cross-validated encoding models of recorded ERPs using pre-trained ConvNet features.

is optimized to process the essence of vast amounts of information, i.e., the gist of a scene, at a high temporal resolution (Biederman et al., 1974; Potter, 1975; 1976; Thorpe et al., 1996; Greene & Oliva, 2008; Sharan et al., 2009).

While differential sampling between the fovea and the periphery is theoretically implementable in ConvNets, the current strategy of ConvNets of uniformly sampling the input by reusing features solves tasks like object recognition a lot more efficiently and can lead to good predictions of higher visual cortex (Yamins et al., 2014). However, enhanced task efficiency does not necessarily reflect enhanced brain alignment (Konkle et al., 2022; Kubilius et al., 2019), emphasizing that the brain solves many more tasks under strong energy and anatomy constraints that ConvNets are not subjected to (Dupont et al., 1993; Orban et al., 1996; DiCarlo et al., 2012). The brain therefore needs to carefully select the right amount of information in an efficient manner, e.g., through differential spatial sampling. We hypothesize that augmenting ConvNets with differential spatial sampling will increase their representational similarity with humans.

The retina is converting photoreceptor signals to a color-opponent signal (Koenderink et al., 1972) that is different from the RGB (red-green-blue) coded input of ConvNets. Adding a color-opponent transformation to the ConvNet input should therefore enhance their biological plausibility and increase neural prediction performance. Furthermore, it is well-known that structures involved in visual processing between the retina (beyond bipolar cells) and the primary visual cortex, such as RGCs (Easter Jr, 1968) or the lateral geniculate nucleus (LGN) (Bonin et al., 2003), process information in a non-linear fashion (Hennig et al., 2002; Macleod et al., 1992; Benardete & Kaplan, 1997). Thus, we add a pre-filter that applies a non-linear contrast-based edgemap-algorithm.

Concretely, we add six pre-filters to a ConvNet that have different levels of biological plausibility. We train these augmented ConvNets on object classification on high resolution images containing a single object. Object images are cropped from high resolution street scene images from a new dataset using manually annotated bounding boxes. We collect human EEG data during visual presentation of these high resolution scene images and evaluate how well the learned ConvNet features are able to predict differences in event-related-potentials (ERPs) across images. We use cross-validated linearized encoding models to map the convolutional features onto the measured EEG data and use

the prediction performances as an indicator for the most successful pre-filter. We present empirical findings for ConvNets augmented with a variety of pre-filter and hypothesize that the degree of biological plausibility will predict the neural prediction performance. Furthermore, leveraging the high temporal resolution of EEG recordings, we are able to investigate differences in temporal processing of visual stimuli. The approach of applying a biologically plausible pre-filter to the ConvNet aims at changing the geometry of the learned representational space and thereby at increasing the representational similarity between humans and ConvNets (Sucholutsky et al., 2023).

## 2 RELATED WORK

**Augmented ConvNets** The approach of augmenting ConvNets with static filters that are not subject to the training data has already led to positive results in increasing representational alignment between humans and ConvNets. ConvNets trained for object recognition augmented with a V1-like layer explain more variance in the primate V1 while simultaneously being more robust to adversarial attacks and other image perturbations (Dapello et al., 2020). This showcases that static, i.e., non-learned filters can enhance the neural prediction performance of ConvNets significantly.

**Cortical Magnification and Foveation** The fact that foveal input has higher information density than peripheral input is reflected in the projections of these retinal regions to the (primary) visual cortex. This effect of cortical magnification, where foveal regions are represented in disproportionately more cortical volume than peripheral regions has been known for decades (Cowey & Rolls, 1974; Levi et al., 1985). Wang et al. (2021) found that this effect simulated by spatial adaptive gaussian blur can be used to enhance biological plausibility of the training of ConvNets while maintaining task performance. Related work by Deza & Konkle (2020) shows that ConvNets with spatial adaptive foveation display increased robustness against image perturbations while maintaining task performance even on compressed information. Together, these findings should be taken as evidence that the highly optimized human retina performs principled computations leading to a robust visual percept that can be adapted for use in computational models.

Moreover, da Costa et al. (2023) propose a method of foveation that transforms regular images into irregularly-spaced images according to the distribution of retinal ganglion cells in the primate retina guided by empirical findings. This method of Ganglion Cell Sampling (GCS) leads to a hierarchy of receptive field properties measured using conventional RF mapping techniques (Dumoulin & Wandell, 2008) that is similar to that of the human visual hierarchy measured using fMRI.

**Peripheral Vision and Scene Perception** The above findings motivate the use of foveation as a method of increasing representational alignment. However, prior studies on human visual perception indicate that peripheral visual processing also plays an important role, especially for scene perception (Rosenholtz et al., 2012), gist processing (Larson & Loschky, 2009), face perception (Park et al., 2023), and even for providing context (Wu et al., 2018; Aldegheri et al., 2023) in object detection, segmentation, or transformation. Wu et al. (2018) show that peripheral information can help improve object recognition performance and robustness to noise in ConvNets. Furthermore, Groen et al. (2013) showed that image-computable scene statistics based on simple pre-cortical filtered inputs (including COC and edgemaps, but also other filters (Scholte et al., 2009; Groen et al., 2012)) are predictive of human EEG data elicited by visual stimulation and allow to disambiguate different models in terms of their predictive power.

## 3 METHODS

### 3.1 DATA COLLECTION

**Subjects** Human EEG data was collected from 31 participants. The institution's Ethical Committee approved the experiment and all participants gave written informed consent before participation and were rewarded with study credits. Four subjects were excluded from analysis because their recordings were incomplete.

**Stimuli / Image dataset** A new dataset of 6130 ultra-high resolution outdoor scenes was used for both human data collection and ConvNet training. Images contain diverse street scenes, have

an original resolution of 5468x3672 pixels, and are stored in an uncompressed format. Each scene was manually annotated with object bounding boxes for a set of 16 common outdoor object classes (see Appendix Table 1). Additionally, a set of 27 indoor scene images were taken to serve as target images in the human experiment.

**Experimental Procedure**   Subjects were presented with outdoor scene images described above downsampled to a resolution of 2155x1440 pixels using a rapid serial visual presentation (RSVP) paradigm (Intraub, 1981; Keysers et al., 2001; Grootswagers et al., 2019). In total, the dataset consisted of 4800 images of which every subject was presented with a subset of 702 images. 1/3 of all images were repeated 10 times throughout the entire experiment while the other 2/3 were repeated 5 times. Resulting EEG data will be averaged across repetitions per unique images to increase the Signal-To-Noise-Ratio (SNR). As commonly done (Gifford et al., 2022), averaging across 5 repetitions yields a low-SNR train dataset to fit the encoding model while averaging across 10 repetitions yields a high-SNR test dataset. Stimuli were shown on a 24-inch monitor with a resolution of 2560x1440 pixels and a refresh rate of 120 Hz. Subjects were seated 63.5 cm from monitor and were using a head rest such that stimulus spanned 50x29.5 degrees of visual angle.

Each subject performed 240 trials. In every trial a series of 20 images of either only outdoor scenes (non-target trial) or 19 outdoor scenes and exactly one indoor scene (target trial) was presented. Participants were asked to perform a target detection task by pressing either 'Y' for a target trial or 'N' for a non-target trial, at the end of the trial. In each trial, subjects were first presented with 3500 ms gray screen and a red fixation cross in the center of the screen. Next, 20 stimulus images were presented for 100 ms and a 300 ms gray screen each. The fixation cross was presented continuously. At the end of the trial, subjects had 3500 ms to respond with a button press and were instructed to blink before the start of the next trial. The session was divided into 8 blocks each followed by an enforced break of at least 30 seconds with self-paced continuation. Stimuli were presented and behavioural data collected using PsychoPy (Peirce et al., 2019).

**EEG Acquisition and Preprocessing**   EEG data was collected using a Biosemi 64-channel ActiveTwo EEG system (Biosemi Instrumentation) with an extended 10–20 layout, modified with two additional occipital electrodes (I1 and I2, while removing electrodes F5 and F6). Eye movements were monitored with electro-oculograms (EOGs) and using an EyeLink100 desktop-mounted system. EEG recording was followed by offline referencing to external electrodes placed on the earlobes. Resulting EEG data was pre-processed using the MNE software package (Gramfort et al., 2013) as follows: high-pass filter at 0.1 Hz (12 dB/octave) and a low-pass filter at 30.0 Hz (24 dB/octave) followed by two notch filters at 50 and 60 Hz; automatic removal of deflections $> 300mV$; epoch segmentation in -100 to 400 ms from stimulus onset; occular correction using the EOG electrodes (Gratton et al., 1983); baseline correction between -100 and 0 ms; and conversion to Current Source Density responses (Perrin et al., 1989). Resulting ERP data was averaged across repetitions per unique image, thus resulting in an sERP specific to each subject, electrode and image.

## 3.2 Computational Modelling

**Model Training**   We trained five alexnet instances (Krizhevsky et al., 2012) for 30 epochs on object classification using object crops as outlined above. Objects were resized to 400x400 pixels, and the model was trained to assign one of 16 classes. The dataset consists of 82.236 objects which are split into non-overlapping sets for training (80%), validation (10%), and testing (10%).

Each model instance is augmented with one pre-filter (see 3.2.1). Importantly, all pre-filters are static, i.e., no parameters depend on the training context. We report the top-1 accuracy of each model on the object crop test set. All models were trained and tested using PyTorch 1.11.0 on a NVIDIA A100 using NVIDIA CUDA 11.3 with cuDNN v8.2.0 support.

### 3.2.1 Image Filters

**Full-Scene**   In the default image condition, models are trained on RGB object crops. During feature extraction full-scene images are used, i.e., images are resized to a resolution of 2155x1440 pixels without cropping. Images are encoded using RGB-encoding.

**Center-Crops**   In the center-crop condition, models are trained on the default RGB object crops. During feature extraction the center-crop filter crops out the central part of the 2155x1440 full-scene images. Here, we included crops of sizes 400x400, 224x224, 112x112, and 64x64 pixels.

**Ganglion Cell Sampling**   We use the Ganglion Cell Sampling (GCS) technique from da Costa et al. (2023) that transforms an image, such that the central part of the image is magnified and magnification decreases non-linearly as a function of increasing eccentricity according to an empirically derived distribution (Watson, 2014) of retinal ganglion cells in the primate retina. We use the default settings reported in table 1 in da Costa et al. (2023) and apply the transform to object crops during training and to full-scenes during feature extraction.

**Color Opponent Channels**   As an alternative color coding to RGB, we use color opponent channels (COC). An RGB-coded image is converted into: intensity , Blue-Yellow-opponent , and the Red-Green-opponent (Koenderink et al., 1972). The COC-transform is applied both during training on object crops and during feature extraction on the full-scene images.

**Edgemaps**   We use an edgemap-algorithm using multiscale contrast magnitude maps from Groen et al. (2013). These edgemaps are originally computed on COC-transformed images but can, however, also be computed directly on the RGB-images. Thus, we treat the COC-transform and the Edgemap-transform as two orthogonal computation, that can be combined or not. The RGB or COC input is first rectified and then normalized. Further, we compute contrast magnitude maps by convolving the image with a static bank of multiscale filters of different sizes (Zhu & Mumford, 1997; Croner & Kaplan, 1995; Ghebreab et al., 2009; Bonin et al., 2005). Subsequently, minimum reliable scale selection is applied (Elder & Zucker, 1998). The edgemap-algorithm is applied both during training on the object crops and during feature extraction on the full-scene images.

### 3.2.2   REGRESSION ON TRIAL-AVERAGED ERPS

**Encoding Model**   We built linearized encoding models to map convolutional features onto the ERP amplitude for each subject, electrode and time point. We used the same pre-filters as applied during training to extract features for the scene stimuli also used during the human experiment. We used the pre-trained features from all max pooling layers (features.2 referred to as 'layer1', features.5 ref. to as 'layer2', features12 ref. to as 'layer3') and from the penultimate fully-connected layer (classifier.5 ref. to as 'feature'). For every model, we included a general encoding model and four specialized encoding model. The general model used features pooled across all four extracted ConvNet layers. The four specialized encoding models only used features from one of the four layers, respectively.

**Model Fitting Procedure**   The stimuli used during the EEG experiment were divided into two sets: a low-SNR set of 5 repetitions per images and a high-SNR set with 10 repetitions. Stimuli from the low-SNR set are used to fit the linearized encoding model (training set) while the stimuli from the high-SNR set are used to evaluate generalizability (test set).

For every model instance, we flattened the extracted features per stimulus in the low-SNR set across all layers and performed a principle component analysis (PCA) across these features. The projections per stimulus of the first 100 PCs then formed the design matrix. The ERP amplitudes for every subject, electrode, and time point was then regressed onto the design matrix. On the extracted features of the stimuli of the high-SNR test set, we applied the same PC projections and then used the fitted encoding model to predict the ERP amplitude for every subject, electrode, and time point. We then quantified each model's predictive performance by calculating the cross-validated Pearson correlation coefficient between predicted and real ERP amplitude.

**Specialized Encoding Model**   For every ConvNet instance we built four more specialized encoding models. For each specialized encoding model we flattened the extracted features from one of the four layers per stimulus in the low-SNR set. The encoding model is then identically built and cross-validated as described in 3.2.2.

**Noise ceiling calculation**   We calculated the noise ceiling, i.e., theoretical maximum of explainable variance of the ERP amplitude across stimuli, per subject, electrode and time point. We split the ERP signal across ten repetitions for the high-SNR set into two non-overlapping splits of 5 repetitions

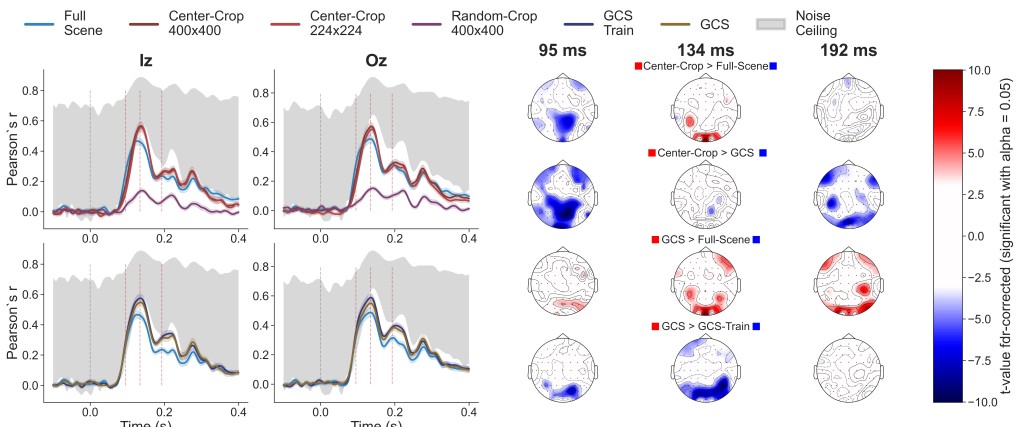

Figure 2: **GCS transform improves EEG prediction performance** (left) Line plots show predictive performance across subjects on high-SNR test set for individual electrodes (first column Iz, second Oz). Colors indicate image-filters used for the feature extraction per model, respectively. Vertical dashed lines show stimulus onset (gray) and time points corresponding to the topoplots on the right. Gray-shaded area indicates the average lower and upper bound of the estimated noise ceiling. (right) Topoplots show t-values for t-tests of the differences of prediction performances between pairs of conditions across subjects at every electrode for three time points. The top row shows the results for the Center-Crop 400x400 condition vs. the Full-Scene condition, second row shows Center-Crop 400x400 vs. GCS condition. Third rows show results for GCS vs. Full-Scene and the last row shows GCS vs. GCS applied already during training. All p-values are fdr-corrected and only significant results with $alpha = 0.05$ are shown in the topoplots.

each. For each subject, electrode and time point, we estimated the lower bound of the noise ceiling as the correlation between the first split of 5 repetitions and the second split of 5 repetitions and the upper bound as the correlation between the first split of 5 repetitions and all 10 repetitions.

## 4 RESULTS

### 4.1 SPATIAL SAMPLING

**Foveal vs. Peripheral Sampling**    Figure 2 shows the average prediction performance across subjects for two posterior visual electrodes using the general encoding model for the RGB model using different spatial sampling. We perform t-tests on pairs of conditions and show the t-values across the whole scalp for selected time points for significant fdr-corrected p-values with $alpha = 0.05$. First, we tested for a difference between the predictions of an RGB-trained alexnet on full-scenes vs. using 400x400 pixel center-crops. We include topoplots for three time points: 95 ms, 134 ms, and 192 ms. We find that the full-scene model yields significantly better predictions at 95 ms than the center-crop model thus predicting ERP data better, earlier in time.Crucially, at the time point of peak performance (134 ms) the center-crop model ($r = 0.56$) outperforms the full-scene model ($r = 0.46$, with $t = 8.63$, $p < 0.0001$ for $\alpha < 0.05$ ) This advantage persists even when cropping to 224x224 pixels (shown in line plots; see appendix Fig. 5 for crops at 112x112 and 64x64 pixels). Any differences between full-scene and center-crop have disappeared at 192 ms and later. Predictions of the same model using randomly centered 400x400 crops (purple line) are significantly worse than all other models post-stimulus (max $r = 0.14$). However, valid predictions can still be found around 134 ms as random crops can overlap with central parts. This comparison shows that drastic differential sampling between foveal and peripheral input to a ConvNet can yield qualitatively different predictions. Peripheral sampling leads to accurate early predictions while foveal sampling yields later but even more accurate predictions.

**Retinal Spatial Sampling**    Above, we showed that including visual features from either the foveal/central part of the image or the periphery helps explain distinct properties of ERP signals

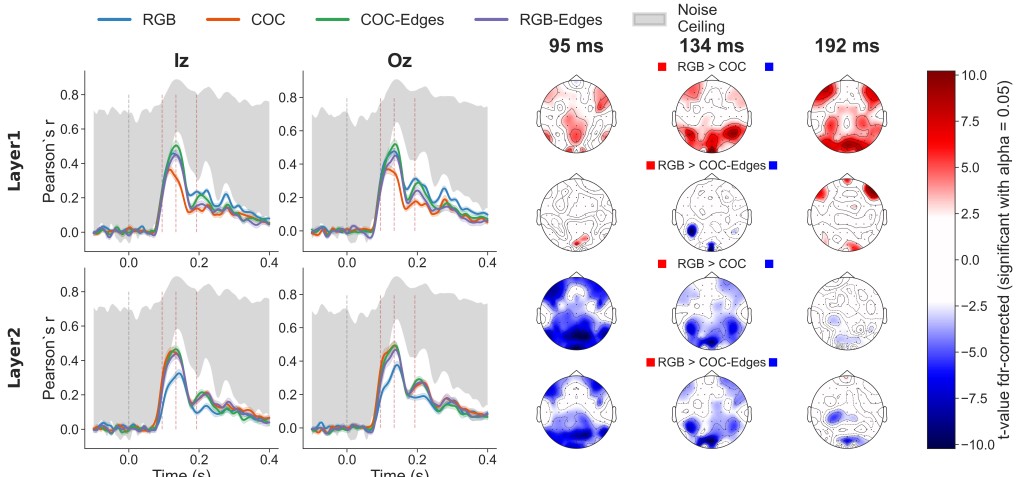

Figure 3: **Improvement color coding is layer-dependent** (left) Line plots show average prediction performances across subjects for specialized encoding model for features from layer1 in the top row and from layer2 in the bottom row on the high-SNR test set for individual electrodes (first column Iz, second Oz). Colors indicate image-filter used for the feature extraction per model, respectively. Vertical dashed lines show stimulus onset (gray) and time points corresponding to the topoplots. Gray-shaded area indicates the average lower and upper bound of the estimated noise ceiling. (right) Topoplots show t-values for t-tests of the differences of prediction performances between pairs of conditions across subjects at every electrode for three time points. The top rows shows results for the RGB vs. COC, second row shows RGB vs. Edgemaps, both for features from layer1. Third row shows RGB vs. COC and fourth row RGB vs. Edgemaps, both for features from layer2. All p-values are fdr-corrected and only significant results with $alpha = 0.05$ are shown in the plots.

elicited by high resolution scene images. Next, we asked whether foveal and peripheral features can be combined more optimally by taking into account the sampling determined by the density distribution of RGCs in the human retina that dictates a non-uniform spatial sampling.

We tested for differences between the center-crop model and the same RGB-trained model tested on GCS-transformed images (Fig. 2 second topoplot row). The GCS model outperformed the center-crop model both early (95 ms) and late (192 ms) in time, but not in between (134 ms). Further, the GCS model outperforms the full-scene model at 134 ms, showing a higher maximum performance ($r = 0.57$, Fig. 2 third topoplot row). Thus, the GCS model simultaneously explains ERP data at early time points as well as yielding maximum prediction results at all later time points.

**Retinal Sampling at Train or Test Time**   So far, we showed that applying the GCS-transform during feature extraction improves prediction performance. This result is thus an effect of a transformation of the input space but not the representational space. Hence, the resulting improvements stems from a input-dependent weighting of visual features and not a training-dependent change of convolutional features. We thus asked, whether applying the GCS-transform already during training would improve predictions even more. We find that indeed, the prediction performance increases for the GCS-trained model for both early (95 ms) and intermediate (134 ms) time points, but not later (see Fig. 2 last topoplot row). FDR-corrected t-tests reveal a significant improvement especially at the intermediate time point for posterior electrodes.

## 4.2   COLOR CODING

We show the results for the specialized encoding models for layer1 and layer2 in Fig. 3 for models using different color coding. The specialized encoding model for layer1 shows that the RGB model (max $r = 0.45$) outperforms the COC model (max $r = 0.31$) at every time point and most strongly in at posterior electrodes ($t = 10.22$ and $p < 0.0001$ for Iz at 134 ms). Moreover, the COC-Edges model significantly outperforms the RGB model at 134 ms at electrodes Iz ($r = 0.50$) and

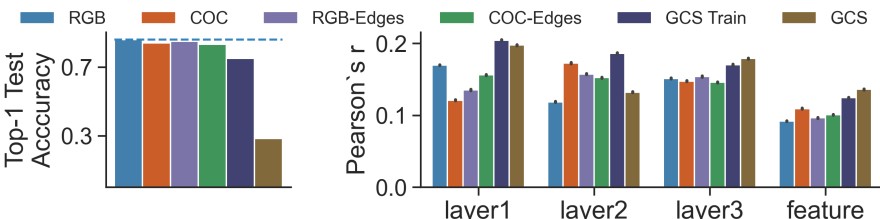

Figure 4: (left) Top-1 accuracy on held-out test set for ConvNets trained using different pre-filter (indicated in colors). (right) Prediction performances for specialized encoding models averaged across all subjects, all 19 posterior electrodes and all time points between 50 ms and 300 ms after stimulus onset. Colors indicate the pre-filter that the ConvNet was trained and tested with (except "GCS": trained on RGB, tested on GCS). Prediction performance decreases for later ConvNet layers and GCS. Advantages of RGB and COC are layer-dependent while Edgemap-trained models yield robust prediction for every layer.

Oz ($r = 0.52$). In turn, the specialized encoding model for layer2 shows that the RGB model is outperformed by all other models. Not only achieve the COC, the RGB-Edges and the COC-Edges models a higher maximum performance but also yield better predictions at early time points.

Using the general encoding model (see appendix Fig. 6), we find the same pattern of results as for layer1. Further, we find no differences between RGB model and RGB-Edges or COC-Edges model at all time points. However, both the RGB-Edges model and the COC-Edges model outperform the COC model at around 130 ms (max $r = 0.45$ for RGB-Edges model and $r = 0.47$ for COC-Edges model for Iz at $t = 0.1344$ s). Overall, we see a clear disadvantage of COC model features for predicting EEG response. The RGB-Edges model performs equally well as the RGB model, showing that the Edgemap-algorithm does not yield better model features compared to RGB. However, the COC-Edges model shows that the Edgemap-algorithm can yield features that recover the prediction performance when compared to the COC model.

To conclude, the RGB model is in most cases the best performing model, together with the edgemap models. For very specific and localized cases we see robust and significant improvements of especially the COC-Edges model using the layer-wise encoding model.

### 4.3 Object Classification Performance

Classification performance of ConvNets decreases with increasing biological plausibility of added pre-filters (see Fig. 4 left). Taking the RGB model as a baseline for (blue dashed line) a drop in test accuracy can be observed for both edges models but mainly for the GCS-trained model. Moreover, applying GCS to an RGB-trained model results in a drastic drop of test accuracy resulting from the distribution shift of the input image.

Furthermore, the specialized encoding models show that later layers generally lead to a decreased ERP amplitude prediction performance than earlier layers (see Fig. 4 right). These layers are commonly associated with a stronger tuning for abstract object classes more than low-level visual features. We therefore see a mismatch between suitability of model features for object classification and for predicting EEG responses.

## 5 Discussion

In this paper, we evaluated methods to increase the representational alignment of brain activity (human EEG) and computational models (ConvNets). ConvNets trained on object recognition are often compared to the human ventral visual stream which is thought to be highly involved in representing objects. However, the comparison assumes that visual processing starts with the cortex, while in biological organisms, substantial processing occurs in peripheral sensory organs and pathways, in this case the eye and the thalamus. Here, we attempted to strengthen the plausibility of using ConvNets as models of human visual processing by augmenting ConvNets with pre-filters that are known to be

computed in the retina and the LGN. We find that differential spatial sampling as an augmentation to ConvNets improves their predictive power while alternative color coding and contrast filters show no or only minimal improvements for EEG predictions.

**Differential sampling**   The human fovea is known to have an effective resolution that is orders of magnitudes larger than that of the periphery. ConvNets sample their input spatially uniform and thus do not approximate the differential sampling of the human retina. The most drastic way of modelling this differential sampling that favours information from the center is to remove peripheral visual information. By using only the a small central part of the input to the ConvNet we show that a large part of the variance in the ERP signal is elicited by foveal stimulation, as randomly taken crops have much lower predictive power. This effect is strongest for very posterior electrodes and highlights the amount of processing that is happening in the primary visual cortex on foveal visual information (cortical magnification). However, the extent of cortical magnification could be overestimated based on these results as a large part of the visual cortex is located in the cerebral fissure and might therefore not contribute to the EEG signal.

Crucially, the high temporal resolution of EEG recordings enables us to distinguish between early and late time windows of ERP responses. We find that ConvNets processing full-scenes can predict earlier parts of the ERPs that are associated with the fast gist processing that is largely dependent on peripheral information. These early predictions are substantially worse for the center-crop model.

Furthermore, the GCS model can account for both, early responses associated with gist processing and for peak prediction performance later in time. Thus, the amount and weighting of information resulting from the GCS-transform seems to be optimal for predicting ERPs elicited by natural scenes.

The spatial invariance of ConvNets might pose a crucial limitation to how similar ConvNets can be to the human visual cortex and how much variance of EEG data they can explain. Here, we see that altering the spatial distribution of the input to a ConvNets improves predictions. We find additional positive effect of including the GCS-transform during training implying a change of the underlying representational geometry across all layers that is beneficial for predicting ERP responses.

**Object Classification Performance**   All models shown in this paper were trained on object classification. Thus, features are not optimized to predict ERP amplitudes and increased prediction performance using the linearized encoding model is a byproduct of the used filters. We see a trend of decreasing object classification accuracy for models using pre-filters with increasing biological plausibility. However, any model that predicts the brain better might also be of risk of showing a decrease in task performance for any single task but might exhibit a broader generalizability.

GCS-trained models show the strongest deficit for object classification. Object crops for training are taken from the full scene by using the smallest rectangle that covers the whole object, often resulting in tightly cropped images. Applying the GCS-transform essentially reduces the information density in the periphery thereby removing important object-related information. Thus, a drop in performance, alongside with increased reliance on local features should be expected. A more optimal way of applying the GCS-transform would be to use it directly on the full-scene (or a larger part of the scene) and thereby emulating the main function of foveation, i.e., to sample an object of interest with a high spatial resolution.

**Color Coding and Contrast Filter**   Augmenting a ConvNet with color or contrast pre-filter has so far not led to major improvements in predicting human EEG recordings. Especially, the linear COC filter is only beneficial after several layers of non-linear processing in the ConvNet. The non-linear contrast filters indeed strengthen this notion as predictions are improved for both early and later layers. However, the default RGB model is the overall best performing model, especially given the computational overhead of the pre-filters. Nonetheless, historically ConvNets have been optimized to process RGB coded input for object classification. It is therefore not surprising that changing the input coding does not yield improvements when compared to an already optimized default. In turn, contrast filtering might yield both task performance improvements as well as neural prediction improvements when trained on task thats e.g., require robustness to noise.

**Future work**   We found a discrepancy of optimal ConvNet features for object classification and for predicting human EEG data. Additional tests for robustness against adversarial attacks or image

perturbation could inform about the benefits of model features that can be used to predict human EEG data. As human vision is highly robust, we would hypothesize that ConvNets augmented with biologically plausible pre-filter exhibit higher degrees of robustness.

To conclude, differential spatial sampling has a strong positive effect on the predictive power of ConvNets for predicting human EEG. While GCS-training improved predictions even further, we also see degraded object recognition performance. Future work should explore applying GCS transforms to model features instead of images. Thereby, the distribution shift of the ConvNet input can be mitigated while keeping the benefit of the GCS transform. Overall, we see differential spatial sampling as an input-transformation as an important asset for neural encoding models.

ACKNOWLEDGMENTS

This work is supported by the Interdisciplinary PhD Programme of University of Amsterdam Data Science Center.

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

## A  APPENDIX

| Class label | Traffic light | SUV | Lamppost | Bench | Scooter | Compact car | Traffic sign | Bike |
|---|---|---|---|---|---|---|---|---|
| # Objects | 3291 | 1705 | 17392 | 2390 | 1843 | 11369 | 11319 | 5546 |

| Class label | Tree | Carrier bike | Truck | Van | Front door | Bollard | Balcony door | Bin | **Total** |
|---|---|---|---|---|---|---|---|---|---|
| # Objects | 1774 | 1780 | 1527 | 10 | 2641 | 8590 | 4428 | 6431 | **82.236** |

Table 1: Object class annotation labels used for ConvNet object classification training. The total amount of object crops used for ConvNet object classification training is 82.236.

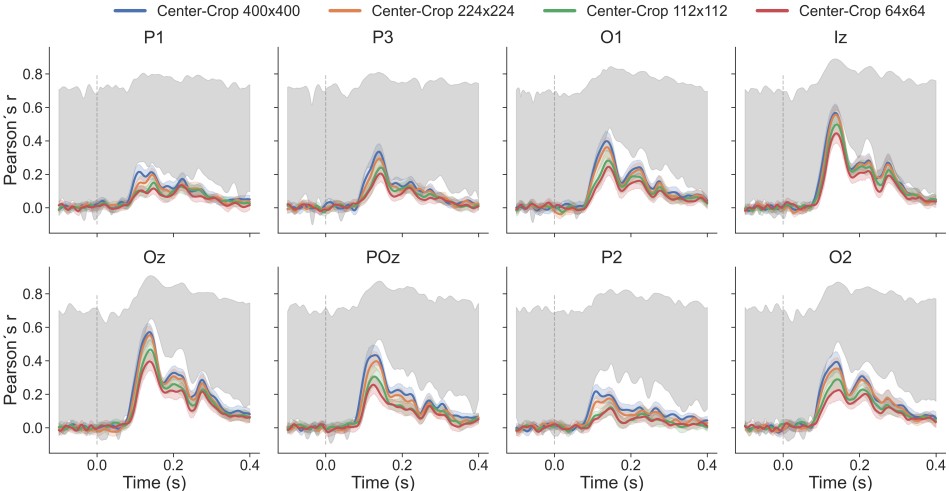

Figure 5: **Information pf foveal image regions is sufficient to yield robust predictions** Plots show average prediction performance across subjects on the high-SNR test set for individual electrodes. Colors indicate size of the center-crop used for feature extraction per model, respectively. All models shown are RGB-trained. Vertical dashed lines show stimulus onset (gray). Gray-shaded area indicates the average lower and upper bound across subjects of the estimated noise ceiling per electrode and time point.

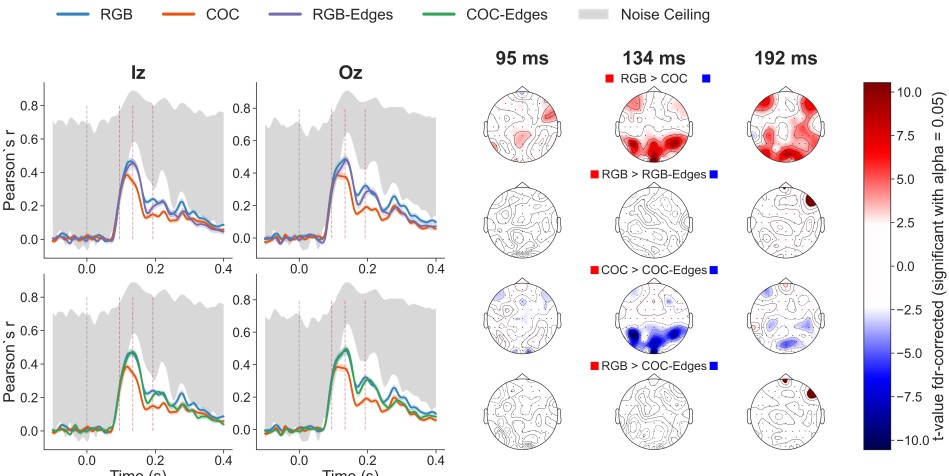

Figure 6: **Advanced color coding or contrast filtering does not improve prediction performance** (left) Line plots show average prediction performances across subjects on the high-SNR test set for individual electrodes (first column Iz, second column Oz). Colors indicate pre-filter used. Vertical dashed lines show stimulus onset (gray) and time points corresponding to the topoplots. Gray-shaded area indicates the average lower and upper bound across subjects of the estimated noise ceiling per electrode and time point. (right) Topoplots show t-values for t-tests of the differences of prediction performances between pairs of conditions across subjects at every electrode for three time points. The top rows shows results for the RGB vs. COC, second row shows RGB vs. RGB-Edges. Third row shows COC vs. COC-Edges and fourth row RGB vs. COC-Edges. All p-values are fdr-corrected and only significant results with $alpha = 0.05$ are shown in the topoplots.

