# OpenReview forum: "Enriching ConvNets with pre-cortical processing enhances alignment with human brain responses"
_ICLR.cc/2024/Workshop/Re-Align — ICLR 2024 Workshop Re-Align Poster_

### Official Review · Reviewer_fKgq · 2024-02-23
**The paper is well-structured and relevant to the theme of Re-Align Workshop of ICLR 2024.**

**Rating:** 3
**Fit:** 3
**Confidence:** 2

**Workshop Review:**

The paper is evaluating the role of six biologically inspired pre-cortical filters on the alignment of the intermediate representations of a deep neural network - AlexNet - with acquired human EEG data. The six filters inspired by the functional and structural properties of retina include: 1) full-scene RGB images, 2) centre-cropped RGB images, 3) Ganglion spatial sampling of RGB images as a differential spatial sampling method, 4) converting RGB images to colour-opponent channels, 5) converting RGB images to edge-maps, 6) converting transformed colour-opponent channels to edge-maps. The EEG data is obtained for 31 participants exposed to high-resolution natural scene images composed of 16 common outdoor object classes. For computational modelling, AlexNet has been trained over the same set of stimuli and object classes. Each model is augmented by one of those six pre-filters which are static and don't have any training parameters. Furthermore, the pre-trained features from all max pooling layers and the penultimate fully-connected layer of AlexNet, after some processing, have been used to regress out the ERP amplitudes of EEG data for every subject, electrode, and time point. The final results show that Ganglion spatial sampling of RGB images as a differential spatial sampling and biologically plausible filter, can improve EEG prediction performance, and thus the model-brain alignment. Other filters do not have that much effects. Overall, I think the paper has been clearly written, correct, novel, and interesting to the community. I suggest only two minor improvements. First, it would be great if the authors motivate more - probably considering the biology of the retina - the reasons for choosing those six filters. Second, it would be great if the authors elaborate more on why they choose this amount of stimuli, object classes, and training epochs considering the number of training parameters in AlexNet. Basically, my question is that how they can make sure that they cannot improve the test accuracy of the AlexNet by increasing training data and epoch without overfitting. There is also one small typo in line 25 of page 9 of the paper. I think "ny" should be written as "any".

**Reason For Not Giving Higher Score:**

N/A

**Reason For Not Giving Lower Score:**

As I mentioned, I think the paper is well-written, clear, correct, novel, and interesting to the community.

**Reviewer Domain:**

machine learning

---

### Official Review · Reviewer_8fNm · 2024-02-23
**Submission53 presents interesting initial findings that could be supplemented with more integration into relevant work and more experiments**

**Rating:** 2
**Fit:** 3
**Confidence:** 3

**Workshop Review:**

The authors try to implement more biologically plausible CNNs to match neural human EEG data by drawing inspiration from the retina. They compare their methodology to previous work and show increase in performance to several previous methods. While the authors already do a pretty good job of centering their work in current literature, there are some missed opportunities. In addition, a few more experiments and interpretative analysis would bring the paper to the next level of interest. Lastly, some changes to the formatting of the paper would aid greatly in clarity.

1) The authors cite the presence of the fovea within the retina as an inspiration for more biologically plausible networks. The fovea of the retina is critical, but other retinal properties also lend to the initial processing of the retina feeding to the LGN, such as the diverse roles of amacrine cells and bipolar cells in complex computations (beyond just edge detection).

2. It is not entirely clear to me exactly which methodologies the authors are applying are novel, and if none, are they simply comparing to on a novel dataset.

3. While training a neural network to predict neural activity is interesting, good performance on this does not necessarily indicate representational alignment. Instead, is there a way to characterize alignment of the network with the visual task-trained (object recognition) networks?

4. Combinations of filters (RGC and color, etc) may be interesting to add into the mix.

5. It would be interesting to probe further into how RGC and edge-detection are truly impacting the activity and representations of the networks through perturbation experiments or otherwise.

6. Please revise the captions of the figures for clarity and consider using sub-panels (A, B, C). Please order the references alphabetically by last name. It's hard to see the differences in Figure 2, I am not convinced the effects are meaningful.

7. Please consider checking out the work of the following labs for deep models of visual systems: Stephen Baccus (Stanford), Dan Yamins (Stanford), Nancy Kanwisher (MIT)

**Reason For Not Giving Higher Score:**

While the findings are interesting, they are not complete enough for a talk.

**Reason For Not Giving Lower Score:**

The findings are interesting and should definitely be presented to the wider community.

**Reviewer Domain:**

neuroscience

---

> ### Author Response · Authors · 2024-04-30
>
> We thank the reviewer for the comments and for pointers and references for further improvements!
>
> Regarding comment 3, we want to emphasise that our method indeed trains ConvNets to perform object recognition and not to directly predict neural data. We then build a linear encoding model, that performs a linear regression on the human EEG data using the pre-trained features from the ConvNet resulting in behavioural alignment (ConvNet can classify objects) and representational alignment (ConvNet features can be linearly mapped onto human EEG data).
> This is indeed the suggested way of characterising alignment using the visual task-trained network.

---

### Official Review · Reviewer_g5Kp · 2024-02-24
**Retinal sampling grids improve predictivity of Human EEG responses**

**Rating:** 2
**Fit:** 3
**Confidence:** 3

**Workshop Review:**

This paper explores the impact of several biologically motivated preprocessing strategies on the ability of ANNs to predict EEG responses of humans in response to natural scenes.

Strengths:
- Several of the applied interventions yield improved predictability.
- The strength of the measurement modality (EEG) is leveraged by examining the timecourse of predictivity. I found the full-view vs. center-cropped result (where full view better predicted early onset responses to be particularly interesting.
- The EEG dataset collected seems like a valuable contribution on its own.

Weaknesses:
- Most of the effect sizes seem to be quite small. I.e. in figure 2 there are a range of sites where there are statisticlally significant differences between settings (as visualilzed in the top plots), but the absolute differences in predictivity seem marginal when examining the left two panels.
- I am skeptical about the setting where GSC is applied during feature extraction to networks trained without this preprocessing step. This would seem to me to put the inputs "out of distribution" from the network's perspective and I am not sure I would expect the extracted representations to be very meaningful. The authors could check this by evaluating classification performance for the vanilla trained model when given GSC processed inputs.
- I am less familiar with the idea that edgemaps are comparable to a precortical processing step. I think the motivation for its inclusion in this study could be made more clear.

Questions:
- In figure 4 GCS is missing from the right panel, or am I missing something?

**Reason For Not Giving Higher Score:**

Effect sizes seem to be small, and I feel that exploring the impact of the interventions in other modalities (i.e. in neural recordings in macaque) would have strengthened the work.

**Reason For Not Giving Lower Score:**

The topic is very well aligned with the workshop, a new dataset was collected, and the experiments seem rigorous and thorough.

**Reviewer Domain:**

neuroscience

---

### Official Review · Reviewer_abvE · 2024-02-24
**This study investigates the incorporation of functional and structural properties of the human retina into ConvNets to enhance their alignment with human brain activity. Their work used foveated stimuli and implementing differential spatial sampling in ConvNets, showing improvements in predicting human EEG data. However, no evidence found that incorporating color and contrast filtering of inputs yields enhanced predictions.**

**Rating:** 3
**Fit:** 3
**Confidence:** 2

**Workshop Review:**

1. Clarity: The paper presents its ideas clearly and understandably. However, there are some sections where additional clarification may be beneficial, particularly regarding the implementation of human-inspired pre-filters, to ensure complete comprehension by all readers (especially for the ML community).

2. Correctness: This paper shows strong evidence that foveated stimuli and differential spatial sampling can improve the alignment between ConvNets and human brain activity. Interestingly, the paper also observes some pre-filters such as color and contrast filtering that exist in human perception but do not lead to improvement in the alignment. Further causal analysis is needed to understand what kinds of features contribute to the improvement.

3. Novety: This paper presents a novel and promising way to improve the alignment between the neural networks and the brain response.

**Reason For Not Giving Higher Score:**

I appreciate the approach of implementing a biologically plausible pre-filter on the ConvNet to enhance alignment between neural networks and brain responses. For future work, it might be worthwhile to consider including more instance-level metrics, as they could shed light on specific image examples where the pre-filter proves beneficial. Such insights would be valuable for gaining a deeper understanding of the fundamental components of human visual processing.

**Reason For Not Giving Lower Score:**

This paper demonstrates the enhancement of ConvNets' credibility as models for human visual processing by integrating pre-filters computed in the human retina and the LGN. Its findings resonate well with the workshop's overarching theme, and the insights into convergence and divergence presented herein point toward a timely and important topic for achieving better alignment between humans and machines.

**Reviewer Domain:**

neuroscience

---

### Official Review · Reviewer_Mnkz · 2024-02-28
**Interesting, rigorous and clear**

**Rating:** 3
**Fit:** 3
**Confidence:** 2

**Workshop Review:**

The paper sets out to test the addition of biologically plausible constraint to convolutional neural networks with the aim of investigating the correspondence between performance of ConvNets and human EEG data. For this purpose, the authors use a series of pre-filters added to ConvNets, focusing on non-uniform spatial sampling, color and constrast, inspired by retinal processing. They use linearised encoding models to map convolutional features onto EEG data (ERPs) and use the prediction performance as an indicator for the pre-filter success. The author systematically investigate the prediction performance of the different pre-filters, and show the non-uniform spatial sampling leads to improved prediction of EEG responses at certain time points.
The paper is very clear, demonstrate rigorous methodology, novel contribution to the field and of high interest to the community.
While the papers shows an interesting and important addition to existing literature, several concerns include:
1. While the benchmark for successful performance was predicting differences of EEG responses across, the ERPs themselves are not shown and it is not clear whether there were indeed substantial differences between them.
2. The aim was to better predict EEG responses. Nevertheless, the tasks of the ConvNets (object classification) and humans (target detection) were very different from one another. In fact, the outdoor scenes were non-targets in the EEG experiment, therefore it is not clear what the level of their processing was, even in early areas of the visual cortex. Indeed, while the pre-filters improved prediction of EEG responses, they did not necessarily improve object classification by the ConvNets.
3. As the author noted, a more suitable non-uniform sampling pre-filter could have included varying sampling within the same image (centre/periphery). It would be nice to see the effect of such pre-filter.
4. Because the pre-filters did not result in better object classification, it would be useful to discuss potential solutions for that for future research, and ways in which both EEG predictions and model performance can be improved.

**Reason For Not Giving Higher Score:**

N/A

**Reason For Not Giving Lower Score:**

The research presented in the paper in of high quality, the findings are useful and interesting to the community, and it sets a good direction for future research.

**Reviewer Domain:**

neuroscience

---

### Decision · Program_Chairs · 2024-03-02

Accept (Poster)